# Auxin: Hormonal Signal Required for Seed Development and Dormancy

**DOI:** 10.3390/plants9060705

**Published:** 2020-06-01

**Authors:** Angel J. Matilla

**Affiliations:** Departamento de Biología Funcional (Área Fisiología Vegetal), Facultad de Farmacia, Universidad de Santiago de Compostela, 15782 Santiago de Compostela, Spain; angeljesus.matilla@usc.es; Tel.: +34-981-563-100

**Keywords:** ABA, primary dormancy, ABI3, auxin, YUC, PIN, ARF, endosperm, integuments, AGL62, PRC2

## Abstract

The production of viable seeds is a key event in the life cycle of higher plants. Historically, abscisic acid (ABA) and gibberellin (GAs) were considered the main hormones that regulate seed formation. However, auxin has recently emerged as an essential player that modulates, in conjunction with ABA, different cellular processes involved in seed development as well as the induction, regulation and maintenance of primary dormancy (PD). This review examines and discusses the key role of auxin as a signaling molecule that coordinates seed life. The cellular machinery involved in the synthesis and transport of auxin, as well as their cellular and tissue compartmentalization, is crucial for the development of the endosperm and seed-coat. Thus, auxin is an essential compound involved in integuments development, and its transport from endosperm is regulated by AGAMOUS-LIKE62 (AGL62) whose transcript is specifically expressed in the endosperm. In addition, recent biochemical and genetic evidence supports the involvement of auxins in PD. In this process, the participation of the transcriptional regulator ABA INSENSITIVE3 (ABI3) is critical, revealing a cross-talk between auxin and ABA signaling. Future experimental aimed at advancing knowledge of the role of auxins in seed development and PD are also discussed.

## 1. Introduction

The evolutionary success of higher plants consists of their ability to produce seeds, units responsible for reproduction, dispersal and survival [1]. Synchronized coordination between hormone signaling networks and environmental cues are being required to control these processes. The viable seed is an entity that originates at the end of the development program progression from the fertilized egg and it is constituted of three genetically different compartments [2,3]. That is filial endosperm (3n) and embryo (2n) on the one hand, and maternal seed-coat (2n) on the other [4,5]. The Angiosperm seed development initiates when the paternal and maternal gametes fuse to create the diploid embryo and the triploid endosperm. In most higher plants, the endosperm initially develops as a syncytium, in which nuclear divisions are not followed by cytokinesis. After a specific number of caryokinesis, the endosperm becomes cellularized. However, the mechanism that regulates the transition to cellularization, a critical process in seed development, remains unknown [6]. The endosperm is usually consumed in the dicots during seed development, while it is retained in mature seeds of monocots. However, the developmental process of endosperm is quite highly conserved in plants. The embryo arrest and seed lethality are produced when endosperm cellularization is impaired. Interestingly, the auxin levels need to be tightly controlled to allow the endosperm to cellularize [7]. On the other hand, seed development relies on a strong interdependent control between three respective compartments that constitute it [3,8,9,10]. Therefore, it is not surprising that all molecular events involved in zygotic embryogenesis are tightly coordinated at the genetic and hormonal levels [9,11,12]. The analysis of zygotic embryogenesis has been basically carried out by the characterization of mutants [13,14,15]. Once the seed tissues are completely differentiated, the embryogenic phase ends and begins the maturation phase in which storage compounds (i.e., proteins and lipids) accumulate in the endosperm (monocots) or in cotyledons (eudicots). Throughout maturation, desiccation tolerance is acquired and programmed cell death occurs; finally, the primary seed dormancy is triggered preventing vivipary [1,16,17,18].

The phytohormone abscisic acid (ABA) regulates multiple physiological processes, including seed maturation, embryo morphogenesis and desiccation, stomatal movements, and synthesis of stress proteins and metabolites [19,20]. ABA is the only hormone known to induce, regulate, and maintain the primary seed dormancy. Thus, seeds of ABA-deficient mutants germinate faster than the wild-type, and transgenic plants constitutively expressing the ABA biosynthesis gene maintain deep seed dormancy [1,21]. During seed development, ABA is produced in all seed compartments, as suggested by the spatiotemporal expression of its biosynthesis genes [22,23]. ABA synthesized in the endosperm and then transported to the embryo is involved in the induction of seed dormancy [22,23,24,25]. Likewise, ABA shows an accumulation pattern complementary to the gibberellin (GAs), being the main hormone that inhibits all the processes induced by them [26,27,28]. Regarding ABA signaling, ABA receptors PYRABACTIN RESISTANT/PYRABACTIN RESISTANT-LIKE/REGULATORY COMPONENT OF ABA RECEPTORs (PYR/PYL/RCAR) bind to ABA to remove the repression by PP2Cs (protein phosphatase 2C) of ABA responses [29]. ABA receptors constitute a 14-member family [30]. The PP2C Arabidopsis cluster includes nine members (i.e., ABI1, ABI2, HAB1, HAB2, AHG1, AHG3/AtPP2CA, HAI1, HAI2 and HAI3) which are negative regulators of early ABA signaling [17,31,32,33]. Removal of PP2C repression allows downstream signaling via OST1/SnRK2.6/SnRK2E SNF1-related protein kinases 2 (SnRK2) [32,33,34,35] (Figure 1). The Arabidopsis genome contains 10 members of *SnRK2*; among them, *SRK2D*/*SnRK2.2, SRK2E*/*OST1*/*SnRK2.6* and *SRK2I*/*SnRK2.3* are essential for ABA responses [35]. The phosphorylation of proteins plays a key role in this ABA signaling pathway [36]. Parallel to what is indicated, ABA is also involved in the regulation and the mechanism of action of DELAY OF GERMINATION-1 (DOG1) a heme-binding protein and master regulator of primary dormancy (PD) that acts in concert with ABA to delay germination [1,11,37,38,39]. Thus, PD and germination are regulated by ABA signaling through a DOG1-AHG1 interaction, acting in parallel with PYL/RCAR ABA receptor-dependent regulation [39]. *DOG1* has been identified in A. thaliana as one of the major regulators of natural PD in conjunction with ABA [40,41]. DOG1 is mainly expressed in seeds, in particular in the vascular tissues of the developing embryo [42]. Besides, it was demonstrated that DOG1 function is not strictly limited to seed dormancy, but that it is required for other aspects of seed maturation, in part by attenuating with ABA and ethylene signaling components [37,43]. Recently, it was demonstrated that ethylene signaling controls seed dormancy via DOG1 regulation [37,44].

On the other hand, a series of evidence clearly relates ABA to the mechanism and mode of action of auxins. That is, it seems doubtless that ABA interacts with auxin to regulate various aspects of plant growth and development [45]. Therefore, some evolutionary crosstalk must occur between both plant hormones. However, the study on the participation of auxin in the final part of seed development (e.g., induction, maintenance and loss of PD) is not developed enough yet. The auxin is a signaling molecule that is present across all domains of life, including algal, moss, liverworts, lycophytes and microorganisms [46,47,48]. Tryptophan (L-Trp) serves as a common precursor for IAA synthesis in plants and auxin-producing bacteria (Figure 1). The auxin is synthesized, stored, and inactivated by a multitude of parallel pathways that are all tightly regulated [48]. Regarding the seed, it is now widely accepted that auxin biosynthesis is required for an array of seed developmental processes (e.g., zygotic embryogenesis and endosperm development, among others) [9,49]. High levels of free-auxins and metabolites found during both early (i.e., cell division and expansion) and last phases of seed development (e.g., endosperm cellularization) suggest that auxin has an essential signaling role [9,50,51]. Recent studies have shown that auxin possesses positive effects on seed dormancy, being in conjunction with ABA the second hormone that induces seed dormancy. Thus, Liu et al. (2013) demonstrate, at the molecular level, a role for auxin in seed dormancy through stimulation of ABA signaling, identifying auxin as a promoter of seed dormancy [52]. On the other hand, the auxin also affects seed germination by altering the ABA/GAs ratio [53]. Until now, the role of phytohormones in zygotic embryogenesis mainly refers to the study of eudicots such as Arabidopsis [54]. In order to generate the appropriate response, the auxin polar transport causes its accumulation in specific cellular places. However, very little is known about auxin biosynthesis and homeostasis, polar auxin transport, and response during early embryogenesis in monocots. Interestingly, some of these features involve auxins that seem to be conserved in both monocots and dicots seeds [55,56]. Auxin is perceived by a transient co-receptor complex consisting of a TRANSPORT INHIBITOR1/AUXIN-SIGNALING F-BOX (TIR1/AFB) binding proteins (i.e., a family composed of six members in Arabidopsis) and a transcriptional repressor Aux/IAA protein whose proteosome degradation is crucial for auxin action [48]. Regarding auxin signaling, auxin-inducible genes (AIG) have AUXIN RESPONSE ELEMENTS (AREs) in their promoters, which are bound by dimers of the AUXIN RESPONSE FACTOR (ARF) TFs [57]. ARFs are TFs that regulate the expression of auxin-responsive genes [57]. In the absence of auxin or in the presence of low levels, AIG expression is prevented by the recruitment of Aux/IAA transcriptional repressors to the promoters via their interaction with the ARFs [58]. ARFs and Aux/IAAs are encoded in Arabidopsis by large gene families with 23 and 29 members, respectively. In the presence of high levels of auxin, AUX/IAA becomes ubiquitinated by the action of the multi-protein E3 ubiquitin ligase complex (SCF^TIR1^) and are broken down by the proteosome complex. ARF can then function, often forming ARF-ARF dimmers that allow the AIGs transcription (Figure 1). The auxin-signaling pathway seems to be conserved in land plants [48].

A striking aspect of the above lies in the participation of auxin as a key hormone, in conjunction with ABA, in the regulation of specific phases of seed life. That is why this review provides the progress made in recent years on the contribution of auxin in the fertilization process and zygotic and post-zygotic embryogenesis phases. Given the recent demonstration of auxin involvement in the seed PD process, this update also considers the events that have led to this outstanding discovery.

## 2. Key Role of Auxin in Zygotic Embryogenesis

### 2.1. Spatiotemporal Auxin Production during Early Embryogenesis

Seed development encompasses a set of morphological, physiological, and biochemical changes and can be divided into three main phases: embryogenesis (including cell division and expansion, and the beginning of endosperm and embryo development), seed maturation and desiccation [54,59]. Notably, several patterning processes are controlled by auxins. The organs and tissues involved in embryogenesis require, among other control mechanisms, precise coordination between cell division and cell differentiation. Embryogenesis is initiated by the zygote polarization (i.e., embryo proper and suspensor, both symplastically connected). That is, the first step of embryonic patterning is the establishment of the apical-basal axis, in which asymmetric distribution of auxin mediated by PIN proteins plays a major role. In the zygote of maize and rice, genome activation occurs shortly after fertilization (i.e., 12 h after pollination) [60]. It is noteworthy in maize that this activation coincides with a remarkable up-regulation of a number of auxin-related genes; namely, those involved in auxin biosynthesis and signaling [60]. Although the three main organs that constitute the seed (i.e., seed-coat, endosperm and embryo) exhibit different morphology and functions, they must coordinate their growth in order to achieve the seed viability [61]. Therefore, phytohormones (i.e., auxin, cytokinins-CKs- and GAs) play key roles in the commissioning and maintenance of this strict regulation promoted by developmental program [56]. The evidenced presence of auxin during all seed development phases suggests that this hormone signaling has a consistent and key role throughout seed formation [7,9,55]. Thus, mutants deficient in auxin biosynthesis, transport, and response are defective in embryogenesis [55]. Moreover, it has been shown that auxin regulation of seed development is concentration-dependent [61]. At present, it is well known that auxin has a critical task for the ovule fertilization, subsequent embryogenesis, and determination of the young embryo polarity [54,55,62,63], among other functions. Likewise, seed-produced auxin is of importance for development and growth, and coordination of the three seed constituent organs [56]. As it happens in other plant organs, the auxin distribution in the seed depends on its polar transport [48,49,63]. Thus, auxin plays a critical role in plant growth and development by forming local concentration gradients [61]. Therefore, local auxin synthesis and metabolism, intercellular transport (i.e., AUXIN1/LIKE-AUX1 family of auxin influx carriers (AUX/LAX), PIN-FORMED family of auxin efflux carriers (PIN; with eight members in Arabidopsis and four expressed during embryogenesis), and some members of P-GLYCOPROTEIN/ATP-BINDING CASSETTE B4 (ABCB/PGP) family carriers) and auxin signaling (Figure 1), acting all in connection, will determine the gradient of auxins and tissue patterning [49,56,64,65,66]. Cellular localization of these carriers is indicative of the auxin flow direction, creating thus morphogenic auxin gradients. That is, auxin has a vital role in determining embryo identity and structure (i.e., embryo axis formation and apical-basal pattern formation during embryogenesis).

It seems now indubitable that the biosynthesis of IAA, the main auxin of plants, takes place from L-TrP and indole-3-pyruvic acid (IPA) as the only intermediary [62]. The L-TrP aminotransferases of the Arabidopsis family (TAA; also known as WEAK ETHYLENE INSENSITIVE8-WEI8) and YUCCA (YUC)-type family of flavin-containing monooxygenases seems to act coordinately to control the IAA biosynthesis in Arabidopsis. Recently, the conservation and diversification of TAA and YUCCA functions were highlighted [62]. There are 5 *TAA* and 11 *YUC* gene members identified in the genome of Arabidopsis. The expression patterns of both enzyme families are spatiotemporally regulated during plant development [63,67,68,69]. Mutations affecting *TAA* and *YUC* have been suitable to demonstrate the importance of Trp-dependent IAA biosynthesis during the onset of zygotic embryogenesis [54,70,71]. As a demonstration, *YUC1*,*4*,*10*,*11* redundantly regulate the Arabidopsis embryonic development by modulating auxin biosynthesis at the globular stage [71]. On the other hand, the production of auxins in stamens and gynoecium has been recently reviewed [72]. During the first phases of embryo sac configuration, *TAA1*, *PIN* and *YUC* are expressed in the chalazal region of the ovule primordium, where the integuments later arise [73,74,75]. The reduction in *TAA1* expression results in losses of ovules, whereas the ovules are small and scarce in *pin1* mutants [73,74]. In addition, the embryo sac cellularization is directly dependent on the setting of an auxin gradient inside the sac (i.e., the highest auxin concentration originates synergids, followed by egg cell and finally, the lowest concentrations originate central cell and antipodals) [73]. Therefore, it is likely there are two key roles for auxin during ovule development: induction of embryo sac development, and control of gametophyte cell differentiation and specification [73]. Altogether, auxin seems to be fundamental for ovule development since its onset [73,75].

Given the small size of both pre- and recent-fertilized ovules, the molecular and hormonal processes that take place in them are barely known. To progress in this subject, an auxin signaling sensor named DII-VENUS was developed. Thus, high-resolution spatio-temporal information about hormone distribution and response during plant growth and development was achieved [76]. Later, in addition to the quantification of auxin through the use of antibodies, various reporter genes (e.g., *DR5v2*) have been developed to track auxin transport, level and signaling in different zygotic embryogenic tissues. Together, the auxin knowledge advanced considerably during onset fertilization [77]. As a demonstration, in the ovule of Arabidopsis and other species, DR5rev::GFP was detected in the young ovule primordia, subsequently in the tip of the nucellus and weakly in the funiculus. DR5::GFP signal is weak or undetectable before pollination or in unpollinated controls [55]. After fertilization, which induces an increase of reporter gene about 7 fold, DR5rev::GFP was localized in the integuments adjoining the micropyle and near the chalazal end of the fertilized ovule [55,78,79]. Taking together all these supporting findings [55,56,67,72], it may be pointed out that: (i) pollination leads to increased auxin levels in the fertilized maternal tissues surrounding the embryo and a localized upregulation of auxin response in the embryo attachment region. That is, auxin plays an essential role after fertilization; (ii) in early globular proembryo (8-cell embryo), YUC3,4,9 constitute the auxin biosynthetic machinery in the top suspensor cell (i.e., basal part of the embryo proper) [67], and PIN7 represents the transport machinery that delivers auxin from the suspensor to specify the proembryo (Figure 2). More specifically, PIN7 is suspensor specific and is polarized toward the proembryo, where the auxin response maximum is established [67]; (iii) *TAA1* and *YUC1*, *4* genes are expressed in few apical cells of globular proembryo state (16-cell embryo), constituting another place of auxin accumulation and triggers polarization of the PIN1 proteins, but not auxin signaling. These processes contribute to the specification of the proembryo basal pole; (iv) the findings (ii) and (iii) generate an apical-basal auxin gradient provoked by the polarized localization of the auxin efflux transporters PIN1 in Arabidopsis and maize [55,80] and PIN7 in Arabidopsis [55]; (v) additionally, at a late globular stage, *YUC1* and *YUC4* were expressed in the same embryonic apical area, and *YUC8* was detected closer to the root pole. Mutations in *YUC8* lead to mitotic arrest during female gametophyte development; (vi) loss of function of *TAA1*/*TAR* and *YUC* genes greatly disturbs embryo development [72]. In summary, since *TAA1*/*TAR* and *YUC* have a tightly controlled expression, it constitutes a means of regulating the spatiotemporal auxin production within concrete tissues of the fertilized ovule. Interestingly, despite its different ovule organization, a similar increase in auxin response in fertilized maternal tissues in Arabidopsis and maize was found [55]. This fact suggests an evolutionarily conserved auxin response. On the other hand, CKs promote auxin biosynthesis genes in various organs and the appropriate ratio of auxin/CKs is important for embryo development [81]. Thus, studies shed light on how auxin and CKs interact with each other to promote and maintain the development of the gynoecium [82]. Likewise, it has been reported that auxin signaling directly activates the transcription of the CKs response regulator genes (e.g., ARABIDOPSIS RESPONSE REGULATOR7 (*ARR7*) and *ARR15*) to reduce the CKs response during early embryogenesis [83]. Besides, CKs affect the apical-basal fertilized gynoecium patterning in a similar way to the inhibition of polar auxin transport [84]. Interestingly, CKs accumulate in the proximal region of the ovule primordium in Arabidopsis [85] and exogenous CKs increase *PIN1* expression [86]. This increased *PIN1* expression is reduced in *cytokinins response factor* (*crf*) mutants [87]. Together, given the small number of results to elucidate hormonal control in the onset embryogenesis (i.e., 1-cell, 2,4-cell, octant and dermatogen), it does not seem unfortunate to venture that CKs take an important part in this regulation.

Finally, recent studies in ovules, mainly in Arabidopsis, have shown that *PIN1* expression was observed in the distal nucellus regions, showing polar localization in epidermal cells, which likely coincides with the accumulation of auxin in the ovule tip prior to megasporogenesis [79,88,89]. Besides embryo cells and suspensor, a third source of auxin can be located in funiculus [88]. Interestingly, the Trp-independent IAA biosynthetic pathway, which involves the cytosol-localized indole synthase (INS), is critical for apical-basal pattern formation during early embryogenesis in Arabidopsis [90]. Likewise, through genetic, biochemical, and functional studies, it was recently evidenced that the coordinated action of biosynthetic pathways of IAA-dependent and independent of Trp regulates the zygotic embryogenesis in Arabidopsis [46,71,90]. The auxin production via IPA is preferably involved in embryogenesis and its synthesis initially starts in the suspensor cells (i.e., the uppermost cell), and then transported into the embryo through the efflux regulator PIN7 [55,91]. Although the synthesis and localization of auxins are being widely studied, its regulation is less so. However, recent findings suggest that certain TFs may be linked to auxinic regulation during embryogenesis. Thus, the fact that one of MADS-box TFs, MADS29, a key regulator in endosperm development, is also induced by auxin in *Or*y*za sativa*, suggests some alterations in the auxin level during endosperm development [92].

### 2.2. The Hypophysis and Suspensor Identity Is Auxin-Subordinate

On the globular stage (Figure 2), when the embryogenic cells acquires its identity, the uppermost suspensor cell differentiates into the hypophysis (HP), which generates the progenitors of the quiescent center and columella stem cells, respectively [91]. That is, HP is the founder cell of the root, stem-cell system. The embryo proper plays a critical role in maintaining the identity of suspensor, which have the embryogenic potential to form a second embryo. Therefore, normally quiescent suspensor cells can develop a second embryo when the initial embryo is damaged, or when the auxin response is locally blocked. That is, through a still unknown mechanism, suspensor cells can be reprogrammed to form a second embryo. In addition to mediating HP specification, it was evidenced that auxin is also involved to maintain suspensor cell identity [93]. An auxin response maximum exists in the HP and this auxin accumulation is critical for HP differentiation. Auxin accumulation is generated by polar localization of the auxin efflux transporter PIN1 (localized to the plasma membrane) and polar auxin transport [67]. On the other hand, HP specification is transcriptionally regulated and its asymmetric division requires protein N-terminal acetylation [94]. To summarize the HP specification, MONOPTEROS (MP) activates their downstream targets, including TARGET OF MONOPTEROS 7 (TMO7), in future vascular and ground tissue cells. TMO7 moves from provascular cells to the uppermost suspensor cell. Additionally, MP promotes PIN1 to the uppermost suspensor cell. Here, both auxin responses through ARF9 and other ARFs, and TMO7 are required to specify the HP. In other words, an increase in auxin in the basal cell of the pro-embryo relieves the repression of MP and expression of TMO7, which moves into the suspensor top cell. When the HP divides, high auxin is transported to the basal daughter cell triggering the inhibition of CKs signaling through direct transcriptional activation of ARABIDOPSIS RESPONSE REGULATOR genes, *ARR7* and *ARR15* [95]. It is important to highlight that CKs are required in the apical daughter cell to specify the quiescent center. This partitioning of auxin and CKs signaling is required for the proper specification of the basal daughter cell as columella and the apical daughter cell as the quiescent center. On the other hand, the auxin response components in the pro-embryo and the suspensor are different [93]. Recently, consistent publications have already confirmed the relationship between auxin and the suspensor identity.

Thereby, (i) the embryo proper functions as an inhibitor to suppress the embryogenic potential of suspensor cells and, thus, maintains the suspensor identity during normal embryogenesis. In this process, the auxin is involved [96]. Likewise, the ribosomal gene named *RPL18aB* is responsible for maintaining suspensor cell identity in *A. thaliana* [97]. Thus, in *rpl18aB*, even when the embryo proper and suspensor were connected, the suspensor lost its identity and developed into a multicellular structure. In this process, the polar auxin transport is disturbed in the *rpl18aB* embryo [98]. Lastly, besides demonstrating the importance of auxin homeostasis in the pro-embryo-suspensor complex, Weijers’ group also identified a genetic network involving several basic Helix Loop Helix (bHLH) TFs that mediate auxin action in controlling suspensor development and/or maintenance of its identity [99]. That is, bHLH TFs are involved in the suspensor auxin response. Specifically, bHLH49 appears to be a notable mediator of the auxin-dependent suppression of embryo identity in suspensor cells [99]. Interestingly, the misexpression of *bHLH49* alone induced excess divisions and even the formation of multiple embryo-like structures in suspensors, similar to the effect of inhibition of the auxin response [99]. However, the specific role of this *bHLH49* gene is under study.

### 2.3. Involvement of Auxins in the Coordination of Endosperm-Integuments Development

As described above, the processes of fertilization and post-fertilization of the ovule are the most studied in relation to the auxin attributions during zygotic embryogenesis. However, the knowledge of the auxin itinerary from its source to essential seed organs (i.e., integuments, endosperm and embryo) and auxin distribution in them (i.e., micropylar and chalazal regions, among others), is still far from known in detail. The use of auxin input reporters (e.g., R2D2) or auxin signaling markers (e.g., DR5; see above) is manifested by contributing to clarify both auxin itinerary and distribution. Within this complex puzzle, the more than likely TF AGAMOUS LIKE MADS-box domain (amino acids 6 to 66) protein called AGL62 plays a determinant role since it is at first involved in the development of central cell and/or endosperm [61,100]. That is, AGL62 contributes to endosperm initiation through repressing auxin biosynthesis genes expression. Data from *agl62-2* phenotype (i.e., endosperm cellularized prematurely and retains auxins) and *AGL62* expression, support the insight that AGL62 is active during the syncytial phase by suppressing the expression of genes needed for endosperm cellularization [100]. Interestingly, *agl62* seeds fail to initiate the constitution of integuments, despite the presence of dividing endosperm [100]. Accordingly, AGL62 seems to be essential for the generation of the signal that starts seed-coat development [101]; the development of endosperm and seed-coat is tightly linked to Polycomb Repressive Complex 2 (PRC2) function. Thus, removal of the endosperm inhibits seed-coat development [102]. It is to highlight that in the early stages of endosperm development, auxin appears to highly accumulate at the margin but is relatively low in the center of this organ [9]. This auxin distribution in maize endosperm is disrupted by the presence of the auxin transport inhibitor *N*-1-naphthylphthalamic acid (NPA), resulting in a multilayered aleurone [103]. At present, it is robustly established that endosperm cellularization is triggered by suppression of *AGL62* at the end of the syncytial phase and that this suppression is mediated by the Fertilization Independent Seed (FIS)-PRC2 (i.e., FIS-PCR2) [99,102,104]. Moreover, AGL62 may be regulated directly or indirectly by imprinted genes, of which genes in the FIS-PCR2 complex are the most obvious candidates. Together, after fertilization of the central cell, the endosperm initiates a signal through the action of AGL62, relieving the FIS mediated repression and leading to the differentiation of the ovule integuments into the seed-coat. In Arabidopsis, the FIS-PRC2 plays a central role in mitotic repression of the central cell and endosperm cellularization [104]. However, the pathway by which the FIS-PCR2 complex suppresses *AGL62* expression is currently unknown.

It was suggested that auxin is the putative fertilization signal that coordinates the endosperm and seed-coat development [105]. Besides, auxin has a role in the induction of endosperm proliferation in Arabidopsis showing that auxin levels are sufficient to override FIS-PRC2 suppression of secondary nucleus proliferation. However, it is still unknown how auxin intervenes in the regulation of the FIS-PRC2 complex during fertilization. On the other hand, it has been shown that the female gametophyte cellularization is directly dependent on the establishment of an auxin gradient inside gametophyte, defining thus the fates of the female gametophyte cells [73]. In addition, female gametophyte development requires both localized auxin biosynthesis and auxin import from the sporophytic ovule [106]. However, auxin alone is not sufficient to form a fully differentiated and cellularized endosperm [61,107]. Interestingly, once fertilization has been consolidated, auxin biosynthesis in the endosperm drives its own development and is responsible for starting the appearance of the seed-coat [54]. Likewise, through a poorly known mechanism, auxin is transported into the integuments to support seed-coat growth [105]. Lastly, auxin exerts its early physiological task directly in the zone where the integuments are beginning their formation [61]. Conversely, the auxin transport from seed-coat to endosperm was also found [51].

To conclude, even though the mechanism of auxin transport between sporophytic and embryonic tissues remains elusive, very recent results further consolidate the facts indicated above. Thus, the expression of *TAA1*, *PIN3* and other auxin transport proteins increases in integuments following fertilization, while the auxin signaling is very significant in both micropyle and chalazal sections of integuments [56]; to confirm, these features are prevented when auxinic transport is hindered [108]. Together, the presence of auxin within the integuments initiates its differentiation into the seed-coat. On the other hand, it was recently shown that fertilization causes restriction of auxin export through funiculus, resulting in the spread of auxin throughout the integument [108]. On the other hand, increased auxin biosynthesis in the endosperm prevents its cellularization, leading to seed arrest [19]; these results suggest that auxin determines the timing of endosperm cellularization. Finally, an epigenetic regulation signaling pathway activates auxin production in the endosperm and its transport from the endosperm to the integuments. This transport requires ABCB/PGP10 auxin efflux proteins in the endosperm, transcriptionally regulated by AGL62 [56]. Together, the data known to date suggest that auxin is specified for regulating embryo, endosperm, and seed-coat development.

## 3. The Auxin-Mediated Seed Dormancy and Auxin-ABA Relationship

Seed dormancy, conceptualized as the incapacity of a viable seed to complete germination despite the conditions are favorable, guarantees that the seed germinates at a suitable time [16]. Dormancy is hormonally induced, maintained and strictly regulated by the modulation of suitable hormonal signaling networks [1,11]. Abscisic acid (ABA) is the hormone known to regulate the induction and maintain PD hindering pre-harvest sprouting (i.e., viviparism) [28,31]. However, at the beginning of this century, a series of studies have committed auxin in seed maturation, PD and germination. Among other findings, earlier studies showed that application of auxin inhibited pre-harvest sprouting in wheat and promoted key tasks in equilibrating PD and seed germination rates [109]. Likewise, it was also demonstrated that ABA represses the Arabidopsis embryonic axis growth during seed germination by enhancing auxin transport and signaling and repressing the level of expression of *AXR2*/*IAA7* and possibly also *AXR3*/*IAA17* [110]. These authors, by using an auxin transport-defective mutant (*aux1-301*), show an accelerated seed to seedling transition in the presence of ABA. This experimentation interestingly points to the fact that an auxin-ABA synergistic interaction takes place in plant growth and development. That is, deficiencies in the auxin signaling pathway result in ABA modified sensitivity during seed germination. On the other hand, it is also confirmed that seed after-ripening is associated with decreased seed sensitivity to auxin [111] and that auxin signaling pathway is activated in parallel to the acquisition of seed longevity [112]. Interestingly, some genetic evidence has suggested the involvement of auxin in the maintenance of PD in Arabidopsis. For example, the reduced PD in *taa1* and *yuc1yuc6* mutants is linked to decreased ABA sensitivity [52,112]. On the other hand, the IAA level in mature seeds appeared to be linked to PD, since those mutants that have reduced IAA contents also show a reduced PD phenotype [112]. However, the mechanism by which auxin controls seed dormancy is a question not yet clearly resolved at the molecular level. Parallel, strong genetic evidence supports a model whereby ABA-mediated inhibition of seed germination requires intact auxin biosynthesis, transport, and signaling. Two notable lines of evidence follow. The first involves ABI3, a TF involved in the initiation and maintenance of the maturation phase and considered to be a major downstream component of ABA signaling. ABI3 is induced by auxin [113,114]. Though genetic and biochemical evidence has shown that ABI3 is required for auxin-activated seed dormancy [52]. Parallel, it was also demonstrated that seeds of the Arabidopsis *abi4* and *abi5* mutants are insensitive to auxin treatment during germination, indicating that ABI4 and ABI5 are important regulators of auxin-mediated inhibition of seed germination [115]. The second evidence indicates that auxin promotes PD and inhibits germination by enhancing ABA action, but together auxin and ABA act synergistically to inhibit seed germination and the auxin-mediated inhibition of seed germination is dependent on ABA [52,116]. That is, it demonstrates in Arabidopsis a molecular link through which auxin activates ABA signaling to inhibit seed germination [52]. To demonstrate the existence of this link, several facts were proven: (i) *YUC1*, *YUC2*, and *YUC6* expression peaks during the later stages of seed development; consequently, it is more than likely that the auxin biosynthesis enhances during seed maturation; recently, *yuc1yuc6* mutant displays a significantly decreased level of PD and premature germination [112]; (ii) ABA function in seed germination is largely dependent on the TIR1/AFB-AUX/IAA-ARF–mediated auxin signaling pathway; (iii) the enhancement of PD by auxin and the inhibition of seed germination by the ABA is dependent on the function of ABI3 whose transcripts are high in dormant seeds but low after germination. At the evolutionary level, it is interesting that the auxin regulatory mechanism evidenced by the He’s group [52] was later found and conserved in liverworts. This last work demonstrates that endogenous auxin works as a positive regulator of liverworts *Marchantia polymorpha* gemmae dormancy. Summarizing, all these shown facts clearly reveal an unequivocal positive correlation between auxin content/signaling and PD, as it was previously demonstrated also for ABA [117].

In considering the above evidence, it is clearly concluded that the manner in which auxin acts on PD is far from known. On the other hand, it is well-founded that during seed germination, the microRNA miR160 is involved in the regulation of auxin-ABA crosstalk reducing thus the ABA effect [118]. This work demonstrates that miR160 negatively regulates ARF10 and the ABA hypersensitivity of *mARF10* (i.e., miR160-resistant form of ARF10) mutant seeds was mimicked in wild-type plants by exogenous auxin [118]. In relation to this last finding, when auxin levels are high, the auxin-responsive TFs ARF10 and ARF16 (i.e., positive regulators) which are targeted by miR160, indirectly promote the ABI3 transcription, and consequently maintain PD levels and repress germination. In other words, since ARF10 and ARF16 likely do not directly bind to the ABI3 promoter [52], they may recruit or activate an additional seed-specific TF(s) to stimulate *ABI3* expression. Conversely, at low auxin levels, ARF10 and ARF16 are repressed by AXR2/AXR3 [110]. Genetic evidence indicates that arf10 and arf1 mutants display resistance to ABA in germination assays, whereas those defective in the transcriptional suppressor ARF2 (i.e., arf2) display hypersensitivity [52,119]. That is, ARF2 is a negative regulator in ABA-mediated seed germination [119]. Lately, it was demonstrated that Germostatin Resistance Locus 1 (GSR1), encoding a tandem plant homeodomain (PHD) finger protein, forms a co-repressor with ARF16 to regulate seed germination. GSR1 physically interacts with ARF16 to possibly make up an unknown still complex functioning in auxin signaling to regulate gene transcription. This compelling finding indicates that GSR1 may be a member of an auxin-mediated seed germination genetic network [120]. In this thorough work and through chemical–genetic screenings was demonstrated that the germostatine (GS) is a small non-auxin molecule that mimics the effects of auxin and inhibits seed germination and specifically acts on auxin-mediated seed germination [120]. Given the notable characteristics of GS, its research has great prospects at both the genetic and molecular levels. Recently, a preliminary, but a striking study in barley provided evidence for miR393-mediated regulation of auxin response and its interaction with the ABA and GAs pathways during seed development and germination [121,122]. Finally, further screening of dormancy mutants is needed to identify the missing link in the ARF10/ARF16–ABI3 signaling cascade.

Overall, all the research done so far on the auxin-ABA interrelationship opens up a lot of objectives, all of which are attractive. Some of them are included in the following section. However, does auxin have any effect on ABA synthesis and GA biosynthesis/signaling pathways? If so, how does this affect take place? A comprehensive analysis of the auxin responsiveness of ABA biosynthesis, transport, and signaling mutants will be required to determine whether ABA acts downstream in any auxin-regulated process and it should be reinforced through genetic and molecular approaches.

## 4. Future Perspectives

The main factors involved in the induction and maintenance of PD have presumably been described. Thus, at the end of the 20th century and in these last two decades, a lot of experimentation was done to try to understand and explain the key role of ABA in the PD process. Evidence accumulated so far indicates that a concerted action of endogenous signals and environmental cues is required for PD to manifest at the end of seed development. Whereby, it is necessary to continue scrutinizing to know how endogenous and exogenous cellular signals regulate the work of the ABA. Therefore, the identification of major genetic and molecular factors is being investigated in detail during seed development, seed storage, and germination. During the study of auxin involvement in plant immunity, it was evidenced that auxin protects and strictly regulates PD through enhancing ABA signal transduction, identifying auxin as a promoter of PD [51,52]. These findings were supported, among other consistent experimentations, by the dormancy variation among seeds with altered auxin synthesis genes. Given that L-Trp-independent auxin biosynthesis contributes to the development of embryogenesis in Arabidopsis, it will be of great interest to investigate its possible participation in PD and its relation with ABA signaling. In addition, the signaling pathway linking auxin/ABI3/PD was hypothesized to be a consequence of the recruitment of ARF10/ARF16 to control the *ABI3* expression. A lot of research will be essential to identify the IAA/Aux/ARF combination that leads specifically to the activation of ABI3. After the very interesting results of Z.H. He’s group, a large number of questions are emerging. Thus, do the same signals affect auxin synthesis and signaling to regulate PD? Further expanding the question, do ABA and auxin affect the homeostasis of other hormones? Regarding this matter, it will be interesting to investigate how *YUC* genes (e.g., *YUC1*, *YUC2*, *YUC6*) are regulated to fine-tune auxin biosynthesis during seed maturation. In other words, a lot of data is lacking at the molecular level to have a coherent understanding of the interaction between ABA/auxin biosynthetic pathways. However, a fact seems shown, ABA induces the synthesis of auxin. Therefore, it will be important to analyze the importance of regulation of hormonal conjugation, degradation, and control of overlapping gene sets. This thorough and complex mechanism for IAA homeostasis is still starting to understand. In addition, one approach of great interest is, can auxin provoke cellular responses that differ according to their cellular concentrations? If so, this fact would give an added value to the determining role of auxin in plant development. On the other hand, recent information demonstrates that auxin acts downstream of ABA to promote a process (e.g., root hair elongation and seed germination). However, it is unknown if there is any process in which the ABA acts downstream of auxin. The study of the possible points of auxin-ABA interaction (e.g., does auxin affect ABI4 and ABI5?) in the regulation of different plant growth and developmental processes is still in its infancy. If so, this fact would give an added value to the determining role of auxin in plant development. On the other hand, recent information demonstrates that auxin acts downstream of ABA to promote a process (e.g., root hair elongation and seed germination). However, it is unknown if there is any process in which the ABA acts downstream of auxin. The study on the possible points of auxin-ABA interaction (e.g., does auxin affect ABI4 and ABI5?) in the regulation of different plant growth and developmental processes is still in its infancy. As is also the intervention of auxin in the seed after the ripening process.

## Figures and Tables

**Figure 1 plants-09-00705-f001:**
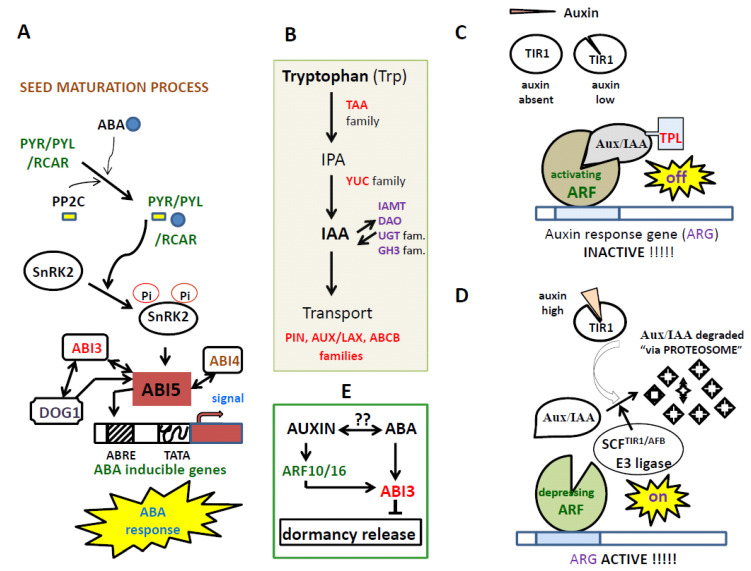
(**A**) In the presence of ABA, the ABA receptors PYR/PYL/RCAR form a complex with PP2C, and this inhibits the phosphatase activity of PP2C and thereby activate SnRK2. The activated SnRK2 subsequently turns on ABRE-binding protein/ABRE-binding factor (AREB/ABF) transcription factors (TFs), which in turn activates the transcription of ABA-responsive genes. Among AREB/ABF TFs, ABA insensitive 5 (ABI5), a member of the basic leucine zipper transcription factor family, plays a central role in regulating ABA-responsive genes in seeds. ABI4 and ABI3, AP2-type and B3-type TFs, respectively, have been reported to function together ABI5 to induce the expression of ABA-responsive genes, and thereby regulate seed dormancy and germination. (**B**) Auxin synthesis and transport involving TAA and YUC enzymes, and PIN, AUX/LAX and ABCB proteins. (**C**) When auxin levels are low, AUX/IAAs prevent ARF regulatory action on auxin-responsive genes. (**D**) If the cellular auxin level is high, auxin promotes interaction between TIR1/AFB and Aux/IAA proteins, resulting in degradation of the Aux/IAAs and the release of ARF repression. (**E**) Hypothesis of interaction between auxin-ABA to inhibit the primary seed dormancy release with ABI3 as a trigger.

**Figure 2 plants-09-00705-f002:**
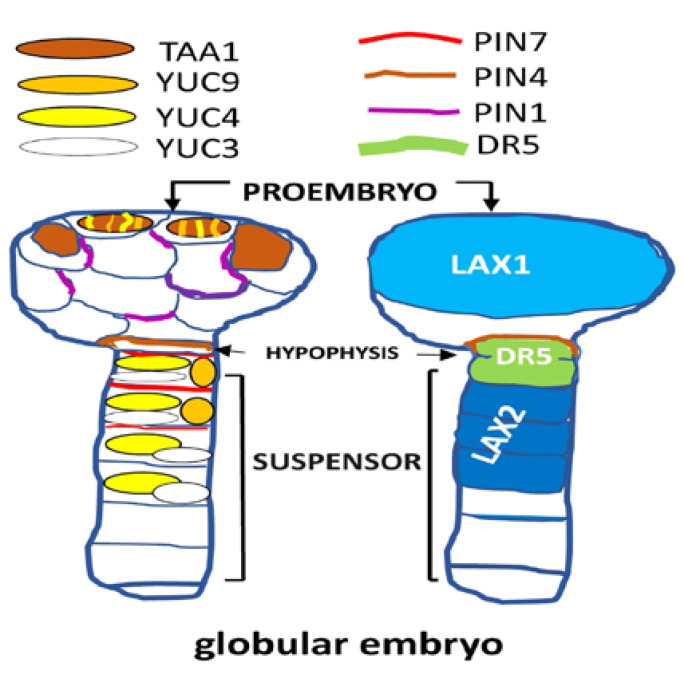
Dynamic of expression and localization (i.e., proembryo, hypophysis and suspensor) corresponding to several genes for the biosynthesis and transport of auxins in Arabidopsis embryos at the globular stage. LIKE-AUX1/2 (LAX1/2); highly active synthetic auxin response element (AuxRE), is referred to as DR5; auxin efflux carrier PIN-FORMED (PIN); YUCCA (YUC); TRYPTOPHAN AMINOTRANSFERASE OF ARABIDOPSIS (TAA1).

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
