# Peer review of "Auxin: Hormonal Signal Required for Seed Development and Dormancy"

_plants, 2020, doi:10.3390/plants9060705_

Round 1
Reviewer 1 Report
This review is well organized and details auxin’s role in seed development and dormancy by combining the newest research findings with more than 60% of references published within the past 5 years.
Some specific comments are listed below:
Line 13: Maybe “together with ABA”?
Line 34: remove “the” in “in the most higher plants”.
Line 46 to 50: considering re-organize this super long sentence.
Line 51 to 80: The whole 2nd paragraph in Introduction is talking about ABA. How about change the focus from only ABA to seed development and dormancy and the related hormones. And then, the third paragraph will move to and focus on Auxin. This change may lead to a more natural flow of the background info to help readers get into the point.
Line 134: Figure 2 is embryo development in Arabidopsis, not seed development. It may not be suitable in here? Besides, this figure was published in Hoove et al., 2015. What is the policy for re-using this figure? Any copyright issues?
Line 158: what is the meaning of “during life seed”?
Line 174 to 175: May be better to change to “Recently, the conservation and diversification of TAA 174 and YUCCA functions were highlighted”?
Line 187: Change “That is” to “Therefore”
Line 201: “gen reporter” should be “reporter gene”?
Line 203: “Taking together the Robert’s updates, your results and other supporting findings”, who is Robert? What did “your results” refer to?
Line 236: “in the species in which the ovule have been examined”, it would be clear to the readers if the author could specify the species at the end of the sentence.
Line 259: “an unknown still mechanism” change to “a still unknown mechanism”.
Line 260-262: considering re-organize the sentence
Line 273: “high auxin is segregated”? Confusing expression. Consider re-organize the sentence.
Line 287 to 289: consider re-organize the sentence, like “Besides demonstrating … complex, Weijer’s group also identified …”
Line 351: change “these authors” to “these results” since only results were cited in the manuscript.
Line 380: the expression of “Parallel, strong genetic evidence by not molecular supports…” is confusing.
Line 405: “Once exposed all of the above, it is clearly concluded the way used by auxin to act on PD is far from known. However, some serious progress have been already revealed”. What could this sentence help with in the context?
Line 437 to 439: Re-organize the sentence into “A comprehensive analysis … process and it should be reinforced through genetic and molecular approaches”.
Line 476: “There is still a lot of work to do.” Seems a very general and casual ending for the manuscript.
Author Response
I appreciate the very good opinion on this draft and the great work of correction and editing carried out by Reviewer 1. All suggested aspects of reorganizing sentences and editing have been resolved. On the other hand, I fully accept the criticism of Fig. 2. Now I realize that it contributes nothing to the content of the text. Therefore, I have replaced it with another one made by myself. It clearly reflects the incidences of different genes in the globular stage (the most studied). All of these genes figure in the main text of this Review. The corrections that appear marked with red bands in the main text were also duly corrected for clarification Finally, I want to justify the only and important reason for me to include the state of the art of the ABA before that of auxins (line 51-80). This decision was very carefully considered. The tittle of this update is: “Auxin: hormonal signal required to seed development and dormancy “. One of the reasons that led me to write this update was the discovery by the He's group that auxin is strongly involved in seed dormancy. Function until then carried out exclusively by the ABA. Currently, a very important part of the mechanism of action of the ABA in the process of seed dormancy is known; and a large number of directly involved mutants and genes have been in deep characterized. However, there are a large number of gaps related to the intervention of hormones and other signaling molecules in the complex puzzle that constitutes the mechanism and mode of action of ABA. This aspect is one of the still current weaknesses of the ABA study. So this update could be of interest to the reader. That said, it does not seem to alter the flow of the Introduction by placing the "master" hormone of seed dormancy (ABA) to the hormone upstart (auxin) in the process.Reviewer 2 Report
Plants-802040-peer-review-v1: Author Matilla
The manuscript of Matilla reviews the evidence for the involvement of the hormone auxin in embryogenesis, its role as a determinant in cellular differentiation and pattern formation in the embryo and in the endosperm and integument development. The author evaluates the sometimes conflicting evidence for the involvement of auxin in the establishment of primary dormancy by highlighting the crosstalk with ABA signaling. The review of the research material supporting the manuscript is objective, clearly indicating the genetic and molecular approaches underlying the conclusions and I do not see any essential omissions or misinterpretations.
There are minor problems in presentation. One is the tendency to report research findings as numerical lists, another is the subjective use of ‘solid’, ‘proven’, ‘for the first time’, let the reader decide as to the value of the literature.
There are many minor errors, typos, missing words etc, that will require meticulous proof-reading. By example:
Line 59 suppress ’induction’
Line 78 replace interfering by attenuating
Line 101 replace proved alterations by metabolites
Line 132 replace to by in
Line 203 delete or rewrite
Line 211 established
Line 214 substitute findings for facts
Line 250 Orzya
Line 259 invert unknown still
Line 296 Substitute Involvement for Implications?Line 334 biosynthesis
Line 345 Rephase …being the…significant
Line 351 timing
Line 382 …notable lines of …
Line 395 decreased level of
Line 398 ABI3 whose
Line 401 define gemma dormancy?
Line 405 In considering the above evidence, it is clearly concluded that the manner in which auxin acts on PD…
Line 448 During the study of
Line 449 delete ’for the…15485)’
Line 451 insert …of PD (51).
Author Response
I appreciate the very good opinion on this draft and the work of editing carried out by Reviewer 2. All aspects of reorganizing sentences, inclusion of references (e.g. [51]) and editing, have been resolved as you suggested. Likewise, words like “proven”, “for the first time” and “solid” were exchanged for less subjetives ones (e.g. shown, demonstrated, etc). Finally, some numbers (references) in square brackets were put in bold.
Reviewer 3 Report
Dear author,
The subject of this paper is of interest to the Journal and scientists and summarized the current state of knowledge since you used literature data from several recent years. The paper provides also interesting questions regarding auxin regulatory mechanisms in seeds that should be answered. This could be helpful also for scientists for designing further experiments.
Some suggested corrections, that might be helpful for the improvement of Ms are below. I found also some editorial errors, so check carefully the text, e.g. dots, etc.
Line 13: should be ‘together with ABA’
Line 17: I suggest ‘essential compound involved in’
Line 20: auxin affects ABA signaling
The introduction started from a long description of ABA involvement is dormancy-related events. However, the title of Ms suggests that the main subject of the paper are auxins. Given the fact that the whole Ms regards mainly auxin involvement in seed dormancy and germination, why did you focus at the beginning on ABA role in these processes? This part should be limited to the most important facts and moved to the section: The Auxin-Mediated Seed Dormancy and Auxin-ABA Relationship. I suggest to start the introduction part with auxins: ‘Auxin is a signalling molecule that is present across all..’
Line 127: In the last part of the Introduction, you suggested:
‘A striking aspect of the above lies in the participation of auxin as a key hormone together ABA in the regulation of specific phases of seed life’
However, only the following statements suggest any potential relationship of IAA and ABA in seed dormancy and germination:
…’ for the first time Liu et al. (2013) demonstrate, at molecular level, a role for auxin in seed dormancy through stimulation of ABA signaling, identifying auxin as a promoter of seed dormancy [52]. On the other hand, the auxin also affects seed germination by altering ABA/GAs ratio [53].’
Thus, based on this and previous comment you should focus on auxins.
Figure 2: doesn’t provide any novel information for the significance of the Ms, which is related to auxins. I suggest choosing several stages of embryo development, in which auxin spatiotemporal distribution was analyzed up to date, and mark this hormone concentration following subsequent developmental stages. It could be more interesting for readers.
In some cases, you bolded citation, e.g. Line 158, while in others Line 165 numbers are non-bolded. Please check it according to the editorial guidelines
Line 195: ‘…because auxin cannot be directly quantified, in planta, by immunolabelling…’ antibodies against auxins are commercially available (provided e.g. by Agrisera)
Line 203: ‘…Taking together the Robert’s updates, your results and other supporting findings…’ – ‘yours results’ it means who?
Line 250: Oriza sativa – Oryza sativa
Line 364: exogenous application of auxin – exogenous auxin or application of auxin. The application is always exogenous.
Line 386: abi4 and abi5 – mutants should be italics
Lina 430: ‘…to answer the question asked at the beginning of this paragraph…’ – which question? It is not clear
Line 435: ‘…any effect on ABA synthesis and and GA biosynthesis/signaling pathways?. If so, how does this effect take place?...’ – remove dots and additional ‘and’
Line 449: (PNAS, 2013, 110, 15485) – remove it and add proper citation number
Author Response
I would like to thank the Reviewer 3 for the constructive feedback, noticeably improving the manuscript.
(1) “The introduction started from.....
I should like to justify the only and important reason for me to include the state of the art of the ABA before that of auxins. This decision was very carefully pondered. The tittle of this update is: é„uxin: hormonal signal required to seed development and dormancy ・ One of the reasons that led me to write this update was the discovery by the He's group that auxin is strongly involved in seed dormancy. Function until then carried out exclusively by the ABA. Currently, a very important part of the mechanism of action of the ABA in the process of seed dormancy is known; and a large number of directly involved mutants and genes have been in deep characterized. However, there are a large number of gaps related to the intervention of hormones (e.g. auxin) and other signaling molecules in the complex puzzle that constitutes the mechanism and mode of action of ABA. This aspect is one of the still current weaknesses of the ABA study. So this update could be of interest to the reader. That said, it does not seem to alter the flow of the Introduction by placing the "master" hormone of seed dormancy (ABA) to the hormone upstart (auxin) in the process.
For a better confirmation of the ABA description before auxins, the paragraph on line 127 was also modified in this sense.
(2) The Figure 2 was deleted and exchanged by a new figure
I totally agree with the criticism about Fig. 2. Now I realize that it contributes nothing to the content of the text. Therefore, I have replaced it with another one made by myself. It clearly reflects the incidences of different genes in the globular stage of Arabidopsis (the most studied). All of these genes figure in the main text of this Review.
(3) With respect to bolded references....
Non-bolded citations were suitably corrected.
(4) Finally, all suggested aspects of reorganizing sentences, editing and the appropriate explanations in various paragraphs have been consistently resolved.
Round 2
Reviewer 1 Report
Thank you for the reply!
Some specific comments regarding the revised version:
Line 51-80: If the author felt it was indeed necessary to have 1/3 of the introduction dedicated to ABA, I would suggest putting some connection sentences between paragraph 2 and 3. We need certain coherency here instead of jumping from ABA (paragraph 2) to auxin (paragraph 3).
Line 127: Suggest change the sentence to “A striking aspect of the above lies in the participation of auxin as a key hormone together with ABA in the regulation of specific phases of seed life” since the highlight here would be auxin.
Line 295-297: “Weijers’s group” could be “Weijers’ group”; besides the citation [98] didn’t match.
Line 355: “was recently shown” should be “it was recently shown”.
Line 467-468: “a large number of questions have and are emerging”, need to remove “have and”, maybe “large” as well since there were only two questions after this statement.
Author Response
The authors have revised regarding the comments. Please find details in the revised file.